# A service evaluation and stakeholder perspectives of an innovative digital minor illness referral service from NHS 111 to community pharmacy

Hamde Nazar[1]*, Cerys Evans[1], Nicole Kyei[1], Laura Lindsey[1], Zachariah Nazar[2], Katie Thomson[3], Andre Yeung[4], Adam Todd[1]

1 School of Pharmacy, Newcastle University, Newcastle upon Tyne, United Kingdom, 2 Department of Clinical Pharmacy and Practice, College of Pharmacy, QU Health, Qatar University, Doha, Qatar, 3 Population Health Sciences Institute, Newcastle University, Newcastle upon Tyne, United Kingdom, 4 NHS England, Local Professional Network–Northumberland, Tyne and Wear, Newcastle-upon-Tyne, United Kingdom

* hamde.nazar@newcastle.ac.uk

**Data Availability Statement:** The data underlying the results is included in the supporting materials

## Abstract

### Introduction

The management of minor conditions represents a significant burden for urgent and emergency care services and reduces the capacity to provide specialist care for higher acuity healthcare need. A pilot Digital Minor Illness Service (DMIRS) was commenced in the North East of England in December 2017 to feasibility test the NHS 111 referral to community pharmacy for patients presenting with minor conditions.

### Objectives

A formative evaluation of the service activity data and qualitative investigation of stakeholders involved in the service design, management, delivery and use, aims to present and investigate the service outcomes.

### Method

Routine service activity data was evaluated during Jan–Dec 2018 to investigate the demographics of patients included in the service; the presenting conditions; and how those referrals were managed by community pharmacies. Semi-structured interviews with NHS 111 call handlers, project team members, community pharmacists and patients were undertaken to investigate the design, management, implementation and delivery of the service.

### Results

13,246 NHS 111 patient calls were referred to community pharmacy during the evaluative period. The most common presenting conditions were acute pain (n = 1144, 8.6%) and cough (n = 887, 6.7%). A large volume of complaints (47.1%, 6233) were resolved in community pharmacy. Stakeholders explained the structured approach to service design,

**Funding:** HN received funds to undertake evaluative work for the Digital Minor Illness Referral Service from NHS England (Grant number: BH181784). The funder provided support in the form of salaries for authors [HN], but did not have any additional role in the study design, data collection and analysis, decision to publish, or preparation of the manuscript. The specific roles of these authors are articulated in the 'author contributions' section

**Competing interests:** AY is employed by NHS England in his role as Chair of the Local Professional Network. This does not alter our adherence to PLOS ONE policies on sharing data and materials

organisation and implementation facilitated successful delivery and management. Patients reported positive experiences with accessing care *via* DMIRS.

## Conclusions

DMIRS demonstrated that patients could be referred to community pharmacy for the management of minor conditions, shifting a burden away from urgent and emergency care. The service data provides key information for further optimisation of service design, and stakeholder training and awareness. The service was acceptable and valued by patients. Evidence from the DMIRS pilot has been utilised to inform recent national healthcare policy and practice around the management of minor conditions within the urgent and emergency care setting.

## Introduction

The NHS is facing unsustainable pressures as evidenced in recent attendance figures for Emergency departments (ED) and General Practice (GP) consultations in England. [1,2] A recent report claimed that 5% of ED consultations and 13% of GP consultations are for low acuity conditions, generally termed minor ailments. [3] These medical complaints are considered common, uncomplicated or self-limiting which do not require medical (i.e. doctor) intervention, but are appropriately managed through symptom relief and self-care. [4]

A recent systematic review investigated minor ailment services in UK and Canada and found structural differences in service design and delivery. [5] However, Paudyal *et al.* have reported that minor ailment services have the potential to reduce GP minor ailment workload, produce positive health outcomes and are associated with positive stakeholder views. [6]

In the UK, the Department of Health has advocated the management of low acuity conditions in community pharmacy, thus freeing up physicians' time and resources in ED departments and GP surgeries. [3] However, although some studies have suggested that pharmacy minor ailment services may be valuable, [7] the services have faced controversy and fluctuating investment in resource and consideration. A recent study found that the main barriers to the embedding and sustainability of these services were the historical poor service design, development and implementation, and lack of recorded evidence. [8] On a more general level, the 2016 independent review of community pharmacy clinical services found the main barriers to successful clinical service provision were:

- poor integration of community pharmacy within the wider NHS, accompanied with a lack of digital inoperability;

- poor recognition by the public and other healthcare professionals as an appropriate healthcare option;

- overly complex and disjointed commissioning and regulatory systems, and

- ineffective use of the diverse skill mix within the workforce. [9]

In order to address the specific challenge of integrating community pharmacy into the wider NHS, NHS England established the Pharmacy Integration Fund (PhIF) to support the development of new pharmacy services, working practices and digital platforms to meet the public's expectations for a modern NHS pharmacy service.

One service piloted by the PhIF was the Digital Minor Illness Referral Service (DMIRS), which tested the technical integration and clinical governance framework for referral from the

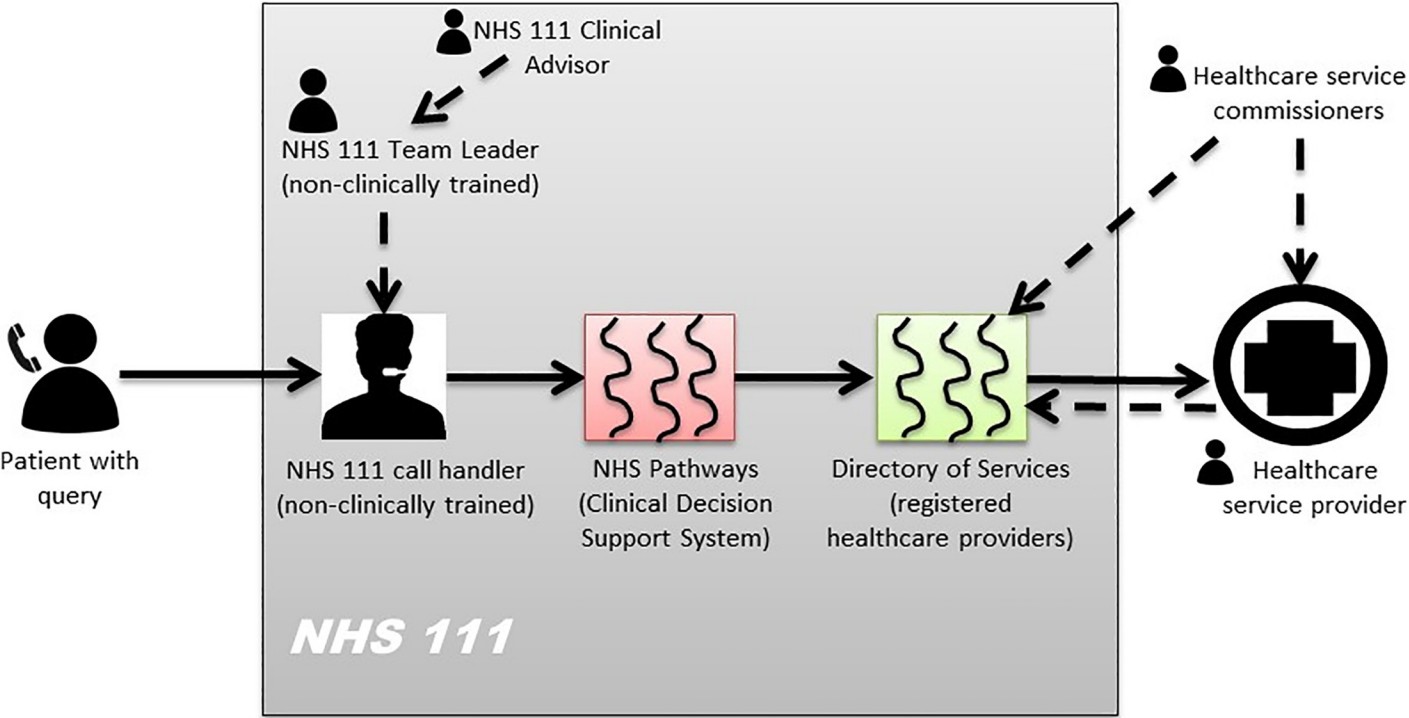

**Fig 1. The NHS 111 system highlighting the key stakeholders involved in the management of a patient call.**

telemedical helpline, NHS 111, to community pharmacy for people needing immediate help with minor ailments. [10] A list of 72 specific minor illnesses, such as cough, colds and sore throat, were identified *via* stakeholder consensus workshops to be safely and appropriately managed in community pharmacy. [10] The workshops were underpinned by participatory service design, with representatives from the key major stakeholder groups in the NHS 111 system, as depicted in Fig 1. [10]

Phase 1 of this pilot involved DMIRS becoming operational across the North East region of England in December 2017.

This study aims to investigate the feasibility and delivery of the DMIRS pilot in the North East of England. Service activity data will be investigated to explore the nature and burden of patient referrals to community pharmacy. The perspectives of service designers, commissioners, providers and users will be captured and scrutinised to assess elements of service implementation and delivery. It is anticipated that these findings are crucial to provide information to service designers, implementers and commissioners about the feasibility in wider adoption of the service but also provide insight on the aspects of the system and process that might require onward optimisation, monitoring and evaluation towards better understanding the outcomes and success of the service.

## Methods

### Study design

A mixed methods convergent design was used, combining the service activity data of DMIRS between Jan 2018 –Dec 2018, and qualitative interviews with stakeholders to evaluate the service. Data collection across these two approaches occurred in parallel and analysis for

integration commenced after data was collected. Findings from this combination of methodologies are anticipated to provide service designers and commissioners with evidence of potential service impact but also information on future service development and optimisation. Ethical approval was obtained from the Faculty of Medical Sciences Ethical Committee, Newcastle University (1612/7270/2018 and 1612/7209/2018).

## Setting

Phase 1 of the DMIRS pilot was commissioned by NHS England Cumbria and North East as a local enhanced service. The service became operational in December 2017 and by June 2018, DMIRS was available in 388 out of 643 pharmacies (who specifically signed up to provide the service) across the North East (NE) region, covering ten clinical commissioning groups (CCGs) across Northumberland, Tyne and Wear, Durham, Darlington and Tees.

## Description of the intervention (DMIRS)

The DMIRS involves the referral of patients from NHS 111, to community pharmacy for specific minor illnesses deemed appropriate for management by a pharmacist.

NHS 111 call handlers, who are not healthcare trained, identify patients presenting with minor illnesses, using a structured questioning approach. The computer decision software prompts the call handlers to refer patients to the closest registered community pharmacy to the patient's home postcode signed up to provide the service from their Directory of Services.

After referral, patients attend the community pharmacy for a private consultation with the pharmacist, structured to investigate the presenting complaint, provide appropriate advice and/or offer an over-the-counter medicinal product to purchase.

The Template for Intervention Description and Replication (TIDieR) checklist, [11] included in the Supporting information (S1 Table), was used to ensure adequate reporting of the service.

## Service activity data

The period of evaluation is Jan 2018 to Dec 2018. No patient identifiable information was included in the service activity data (this has been made available in Supporting information (S1 File)). From NHS 111 the extracted routinely recorded service activity data included:

- Patient demographics (age and gender);

- The shortened postcode of the patient, including their 2015 Index of Multiple Deprivation (IMD) decile, the postcodes of their GP practice and the community pharmacy to which the patient was referred;

From the community pharmacy routine service activity data the following was extracted for purposes of evaluation:

- The time of the NHS 111 call and the time the referral was actioned;

- The minor condition for which the patient called NHS 111;

- The outcomes of the referrals and reasons for rejections where this was provided;

- The completion outcomes of the community pharmacist consultation.

IMD 2015 deciles were used to map the deprivation of those using the service. The IMD is an overall measure of multiple deprivation experienced by people living in an area, and comprises 37 separate indicators organised across seven domains of deprivation (income,

employment, health and disability, education, skills and training, crime, barriers to housing and services and living environment) which are combined, using appropriate weights. [12]

Quantitative data relating to the service activity was analysed using descriptive and. Text entered into free text boxes was coded manually to enable inclusion in the analysis.

## Stakeholder investigation

Semi-structured telephone interviews were used for data collection in the attempt to engage with as many stakeholders as possible causing the minimal inconvenience.

## Sampling

Purposive sampling with the help of the DMIRS project manager [AY], was used to identify the key project team members.

Convenience sampling was used to recruit the other participants. NHS 111 call handlers were invited to take part in the study via an email from the Team leader (a project team member). They were asked to contact the independent researcher [HN] directly about participation. Community pharmacists, who had indicated interest in the evaluation at a DMIRS engagement event, were approached. These stakeholders were considered as key informants, [13] capable of providing key insight into the implementation and delivery of the service. Patients attending the community pharmacy for their DMIRS referral were asked to provide consent and their preferred mode of contact to partake in future service evaluation.

Data collection from all stakeholders continued until inductive thematic saturation was reached whereby no further new codes or themes could be identified during analysis rather than aiming for completeness of existing theoretical categories (theoretical saturation), which would have been a strategy aligning more strictly with a grounded theory approach. [14]

## Recruitment

All stakeholders were contacted by their preferred method. Where emails were not provided in the first instance, these were requested to send the participant information sheets and consent forms electronically for consideration and completion prior to any involvement in the study.

## Data collection and analysis

The Consolidated Framework for Advancing Implementation Research [15] was utilised in developing the key informant interview topic guides. The recommended structure for these topic guides were informed by recommendations from O'Haire *et al*. [13]

For the patient topic guide, questions were mapped to stages of the referral process. Topic guides were tested for face validity with a sample of each of the stakeholder groups and are available in the Supporting information (S3 Topic Guides).

Telephone interviews were audio-recorded with consent, transcribed verbatim and analysed using thematic analysis. An inductive approach to data analysis was adopted, where themes were generated empirically from the data with no imposition of *a priori* presumptions. This constructivist orientation is dependent upon the mutual interpretation of the interviewer and interviewee as the interview unfolds, where the final analysis is assumed to reach a balance between the interacting interpretations. [16] To minimise researcher bias in this process, two researchers [HN, ZN] independently analysed the transcripts, meeting intermittently to discuss impressions and initial codes and categories. Where discrepancies in findings and abstractions occurred, a third member of the research team [LL] was consulted for consensus.

The thick description derived from the interviews, aimed to improve the transferability of the findings and conclusions thereby offering a form of external validity as recommended by Lincoln and Guba. [17] Further incorporating researchers within the team not involved in the research process, allowed the process and product of the study to be scrutinised providing an external audit to increase dependability. [17]

## Results

### Referral data

A total of 13,246 patient calls to NHS 111 were referred to community pharmacy *via* DMIRS during the evaluative period. This equates to approximately 1,103 monthly referrals to the 388 community pharmacies across the pilot region, equating to around 2.8 referrals to each pharmacy over the period. The rate of referrals fluctuated throughout the year, but, generally, the numbers increased during the winter months (Oct-Feb), with an observable dip during the months that followed (Feb-Sept), as shown in Fig 2.

The calls were made across a 24 hour period, with the lowest volume occurring in the early hours after midnight 02:00–05:00 (n = 196, 1.5%), with an increase between 05:00–08:00 (n = 679, 5.1%), with the largest volumes occurring between 08:00–12:00 (n = 4,849, 36.7%), then numbers slowly declined again throughout the day. The increase in referrals for younger

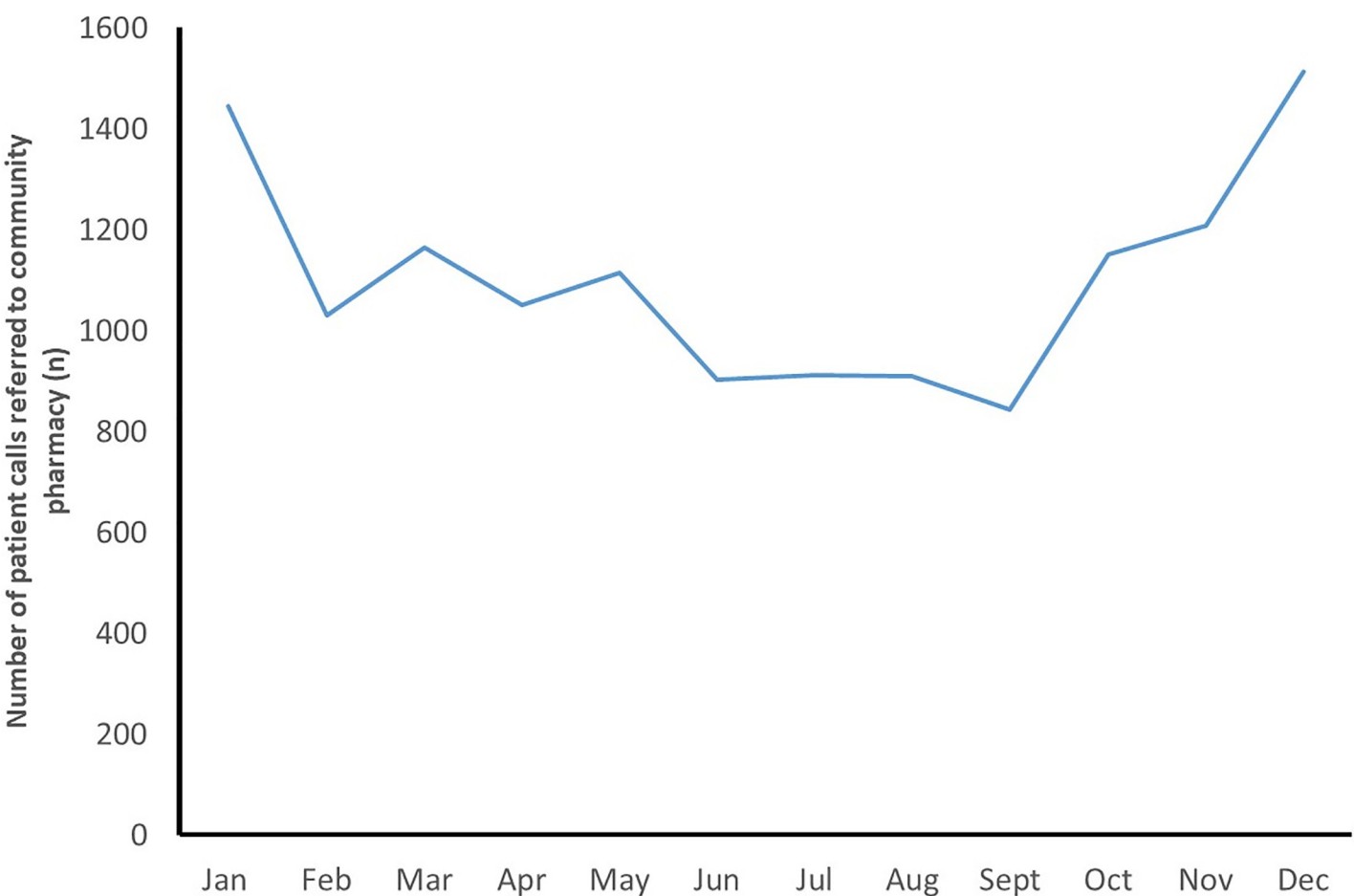

**Fig 2. The number of NHS 111 calls that were referred to community pharmacy over the course of 2018.**

patients (0–9 years old) started earlier in the day in comparison to other age groups; another peak in the afternoon to early night time, 15:00–20:00 was also observed for this patient group.

The largest proportion of referrals, 25.1% (n = 3314) were for patients aged between 0–9 years old, followed by those aged between 20–29 years, representing 21% (n = 2793) of the volume. Decreasing numbers of patients called through to NHS 111 and were referred to pharmacy with increasing age, to the lowest population of 90–100 year olds, representing just 0.2% (n = 20) of the referrals. More referrals were made for females than males (n = 7483, 56.5%, *vs.* n = 5763, 43.5%).

Across the listed conditions, there was some similarity in the minor conditions referred to community pharmacy for both males and females, with acute pain (n = 1144, 8.6%) and cough (n = 887, 6.7%) being the most common (Table 1).

The age profiles for patients presenting with common minor conditions are shown in Fig 3. Allergic rash, cough and high temperature were most commonly observed in younger patients (0–9 years old), while people presenting with acute pain was most common in the 20–38 years age range.

Of the 13,246 calls, 6233 (47.1%) were completed, 2786 (21.0%) escalated, and 4227 (31.9%) rejected (Table 2).

For the most common conditions, the level of completion of the NHS 111 referral to community pharmacy was good (> 60%) (Table 3), with infections being the exception (33.7% completion).

For other less common conditions, completion rates were lower (< 50%), with many falling below 30%, as illustrated in Table 4.

Only 24.2% (n = 1,506) of patients with a completed referral received a sale or supply of an over-the-counter medication. The most commonly supplied medications included: antihistamines (n = 152, 10.7%), non-opioid analgesics (n = 195, 12.9%), non-steroidal anti-inflammatories (n = 208, 13.8), and sympathomimetics, cough suppressants, demulcents and mucolytics (n = 141, 9.4%).

People living in the most deprived areas of England where more commonly referred into the service than people living in more affluent areas. Indeed, around 25% of patients referred to a community pharmacy were from the IMD decile 10 (the most deprived 10% of Lower Level super output (LSOAs) areas nationally), shown in Fig 4. While community pharmacies located in the most deprived areas where responsible for receiving the most patient referrals, as shown in Fig 5.

**Table 1. Common conditions suffered by males and females that were referred to community pharmacy.**

| Condition | Female (n, % of total females) | Male (n, % of total males) | Total (n, %) |
|---|---|---|---|
| Acute pain | 635 (8.5) | 509 (8.8) | 1144 (8.6) |
| Allergic rash | 280 (3.7) | 184 (3.2) | 464 (3.5) |
| Colds | 169 (2.3) | 136 (2.4) | 305 (2.3) |
| Cough | 462 (6.2) | 425 (7.4) | 887 (6.7) |
| Diarrhoea | 142 (1.9) | 122 (2.1) | 264 (2.0) |
| Headache/Migraine | 166 (2.2) | 124 (2.2) | 290 (2.2) |
| High temperature | 180 (2.4) | 175 (3.0) | 355 (2.7) |
| Infection | 172 (2.3) | 184 (3.2) | 356 (2.7) |
| Rash | 163 (2.2) | 122 (2.1) | 285 (2.2) |
| Sore throat | 284 (3.8) | 157 (2.7) | 441 (3.3) |
| Strains/sprains | 143 (1.9) | 125 (2.2) | 268 (2.0) |
| | | | **5059 (38.2)** |

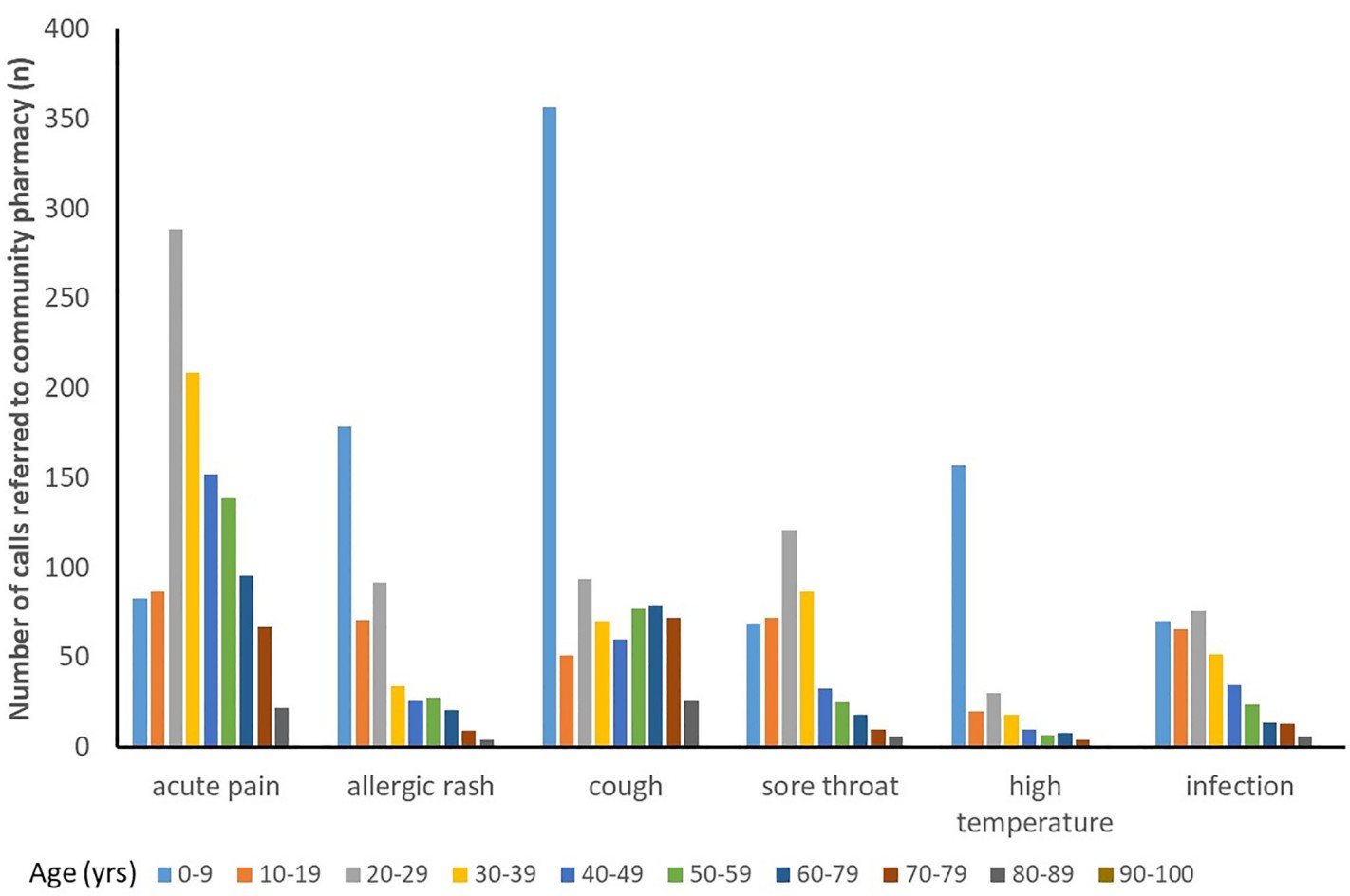

**Fig 3. The age profiles for patients who were referred to the community pharmacy according to the presenting condition.**

## Stakeholder investigation

A mix of stakeholders were interviewed between Dec 2018 and May 2019, including: NHS 111 call handlers *[CH]* (n = 14); Project team members *[TM]* (n = 6); Community Pharmacists *[CP]* (n = 12) and Patients and the Public *[PP]* (n = 30). Average length of interviews was 21.5 mins ± 7.1 with a range 12.3–32.4 mins.

The findings from the stakeholder interviews are presented below with illustrative quotations.

During the analysis, themes emerged from the CH, CP and TM interviewee transcripts that demonstrated how DMIRS had caused aspects of change in thinking and/or process of working, e.g. the intervention had brought about change in the behaviours of NHS 111 call handlers, and insinuated that DMIRS will bring about change in patients seeking care and advice for low acuity conditions. As such, the themes have been mapped to the components of the behaviour change wheel as described by Michie *et al*. [18] The specific aspects of the wheel have been used to frame the emerging themes to facilitate coherency of discourse below and has been visually summarised in Fig 6.

## NHS 111 team members

- Psychological Capability and Reflective Motivation of NHS 111 call handlers enhanced by Education, Enablement & Training

**Table 2. The outcomes of the NHS 111 calls that were referred to community pharmacy.**

| Outcome | Additional notes | Number of patients (%) |
|---------|-----------------|----------------------|
| Completion | Advice given with safety netting* | 3442 (55.2) |
| | Advice given and sale/supply of a medicine | 2776 (44.5) |
| | Appropriate treatment at pharmacy, e.g. wound dressing | 2 (-) |
| | Unknown | 13 (-) |
| Escalation | Pharmacy called 999 | 8 (-) |
| | Referral to GP | 1171 (42.3) |
| | Referral to NHS 111 | 1046 (37.7) |
| | Referral to secondary care | 352 (12.7) |
| | Referral to specialist, e.g. dentist | 28 (1.0) |
| | Referral unspecified | 181 (6.5) |
| Rejection | Did not attend | 355 (8.4) |
| | Inappropriate referral, escalation without contact | 128 (3.0) |
| | Patient has appointment with another healthcare professional | 739 (17.5) |
| | Patient recovering/no longer required help | 389 (9.2) |
| | Other | 33 (0.1) |
| | Patient seen by another healthcare professional | 441 (10.4) |
| | Patient uncontactable | 1894 (44.9) |
| | Unknown | 248 (5.9) |

*safety netting involved providing patients with information about the development of signs and symptoms that would require they seek further medical attention (otherwise termed 'signposting').

All call handlers described and demonstrated commitment to the intended objectives for DMIRS as shifting the burden of low acuity conditions from urgent and emergency care (UEC) to community pharmacy. They were very positive about the progress of the service and described a strengthened belief in community pharmacy as the appropriate healthcare professional to refer those patients. Most had some previous awareness of the potential roles and responsibilities of community pharmacists. To the participants, the dedication of the project manager in delivering a tailored presentation to them in their workplace and coaching them was immeasurably valuable.

> 'When we had that presentation by the pharmacist, he was great. He explained what pharmacists could do. . .he then spent time, one-on-one with us to help us understand pharmacists better and answer our questions.' [CH3]

Gaining this knowledge provided the interviewees with confidence in advising calling patients to attend community pharmacy for their complaints. Call handlers described how their negotiating ability was strengthened by their improved awareness, enabling them to reassure and educate the patients about the abilities of community pharmacists.

> 'I had a lot more to say to the patient about what a pharmacist could do for them. It helps. . .we have training in negotiating, but we don't have a script to read to convince people. . .The visit from the pharmacist built my confidence to offer community pharmacy to the patient.'[CH11]

• Physical Capability of NHS 111 call handlers facilitated by the Environmental Restructuring

**Table 3. The outcomes of the most prevalent conditions that were referred onto community pharmacy.**

| Condition | Outcome (n, %) | | | Total (n) |
| --- | --- | --- | --- | --- |
| | Completed | Escalated | Rejected | |
| Acute pain | 792 (69.3) | 343 (30.0) | 9 (0.8) | 1144 |
| Allergic rash | 387 (83.4) | 72 (15.5) | 5 (1.0) | 464 |
| Colds | 254 (83.3) | 46 (15.1) | 5 (1.6) | 305 |
| Cough | 656 (74.0) | 224 (25.3) | 7 (0.8) | 887 |
| Diarrhoea | 222 (84.0) | 41 (15.5) | 1 (0.4) | 264 |
| Headache/Migraine | 178 (61.4) | 112 (38.6) | 0 (0.0) | 290 |
| High temperature | 245 (69.0) | 108 (30.4) | 2 (0.6) | 355 |
| Infection | 120 (33.7) | 235 (66.0) | 1 (0.3) | 356 |
| Rash | 174 (61.1) | 109 (38.2) | 2 (0.7) | 285 |
| Sore throat | 313 (70.9) | 124 (28.1) | 4 (0.9) | 441 |
| Strains/sprains | 217 (80.9) | 51 (19.0) | 1 (0.4) | 268 |

DMIRS was reported as non-disruptive to the normal daily working practices of the call handlers. The hurdle of recommending it to patients and managing reactions was the most noticeable adjustment to practice. The patient response was largely of acceptance and perceived compliance. Even the occasional declines were something the call handlers were used to.

> 'It's part of what we see...patients declining what we offer them. They know they want to see a doctor and just won't accept anything else...I think the rate of that happening with this [DMIRS] is similar to what we usually see.'[CH4]

Similarly, call handlers articulated that the current referral back to NHS 111 *via* an escalation process from community pharmacy, was not noticeably significant and appeared justified, as patients were often found to have more concerning symptoms when they consulted a pharmacist than they originally disclosed on the phone.

> 'There hasn't been a lot that has come back to us from what I can remember. But the ones that have, have been those that seemed more well when they spoke to us. So it makes sense they were referred back' [CH7]

Call handlers believed this service was adding value to patients and the wider NHS system and advocated for continued investment. There was an expressed appreciation and

**Table 4. The conditions that demonstrated higher rates of escalation after being referred onto community pharmacy.**

| Condition | Outcome (n, %) | | | Total (n) |
| --- | --- | --- | --- | --- |
| | Completed | Escalated | Rejected | |
| Chest infection | 13 (32.5) | 27 (67.5) | 0 (0.0) | 40 |
| Ear condition (other) | 14 (35.0) | 25 (62.5) | 1 (2.5) | 40 |
| Gout | 19 (44.2) | 23 (53.4) | 1 (2.3) | 43 |
| Gynaecological query | 11 (42.3) | 15 (57.7) | 0 (0.0) | 26 |
| Impetigo | 29 (34.1) | 54 (63.5) | 2 (2.4) | 85 |
| Infection (unspecified cause) | 120 (33.7) | 235 (66.0) | 1 (0.3) | 356 |
| Shingles | 47 (46.5) | 54 (53.5) | 0 (0.0) | 101 |
| Tonsillitis | 8 (27.6) | 20 (69.0) | 1 (3.4) | 29 |
| Urinary tract infection | 17 (30.1) | 38 (69.1) | 0 (0.0) | 55 |

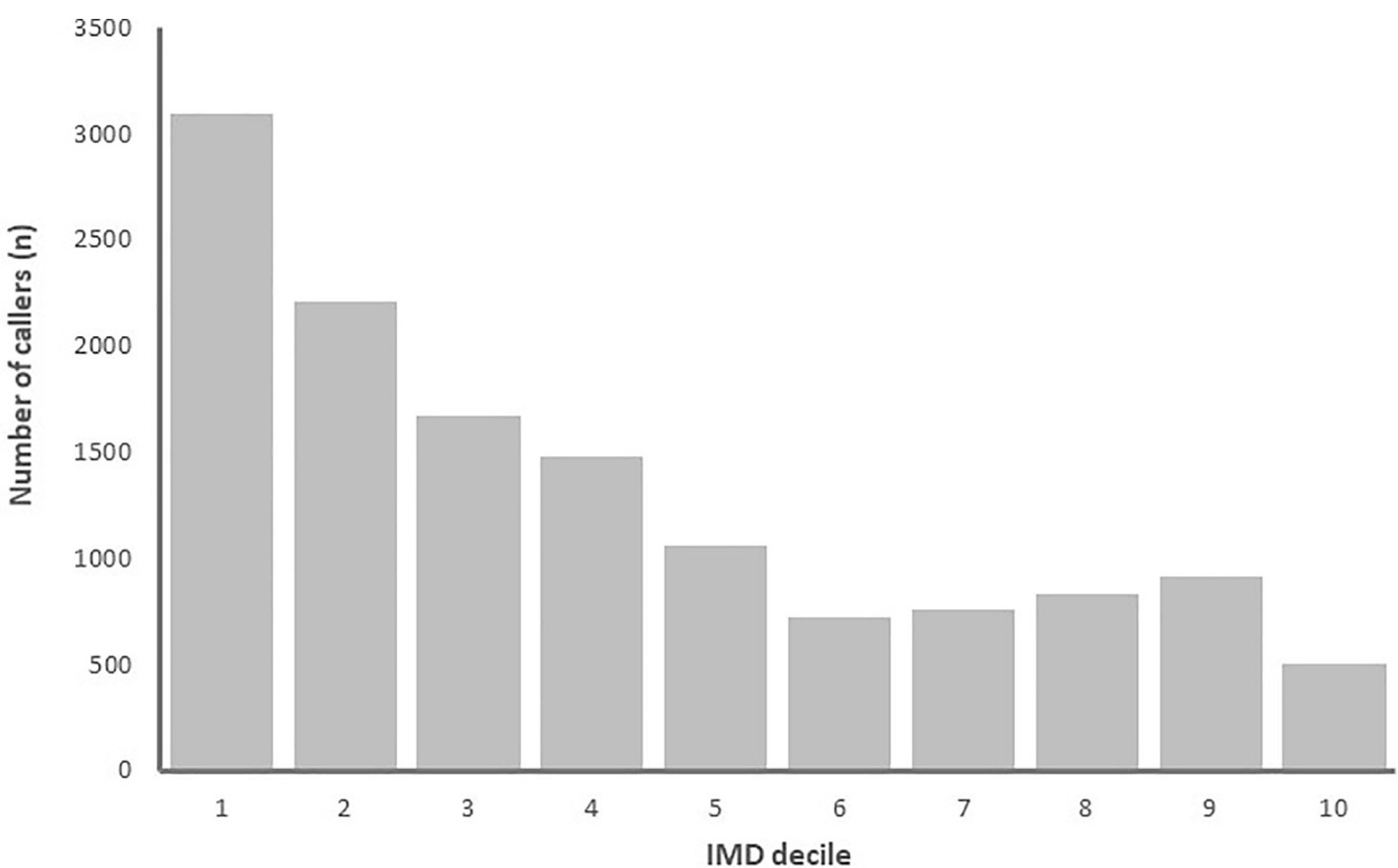

**Fig 4. The number of community pharmacy NHS 111 referrals according to area-level deprivation of the patient (1 is highest deprivation, 10 is the lowest).**

anticipation that the current offering of DMIRS could lead to overall patient and public education about self-care and the possibility of the community pharmacy being the first port-of-call without having to call through to NHS 111.

There were recommendations of wider community pharmacy engagement, more use of the pharmacists' skills with respect to management of minor infections and prescribing of antibiotics specifically.

*'It would be great if we could refer more patients into it* [DMIRS]. *Then there would be more awareness and acceptance. . .So if pharmacists could provide antibiotics or help with more conditions like infections, things would be easier all round.'[CH2]*

*'It's really about educating the public about what pharmacy can do, so that's who they go to next time. . .and tell other people.'[CH13]*

**Project team members.**

- Physical Opportunity & Capability of patients and pharmacists facilitated by Environmental restructuring

For the team members, DMIRS offered means to educate patients and the public, improving their psychological capability and perceived opportunity of accessing care and advice from a community pharmacy for low acuity conditions. They all emphasised how DMIRS was an

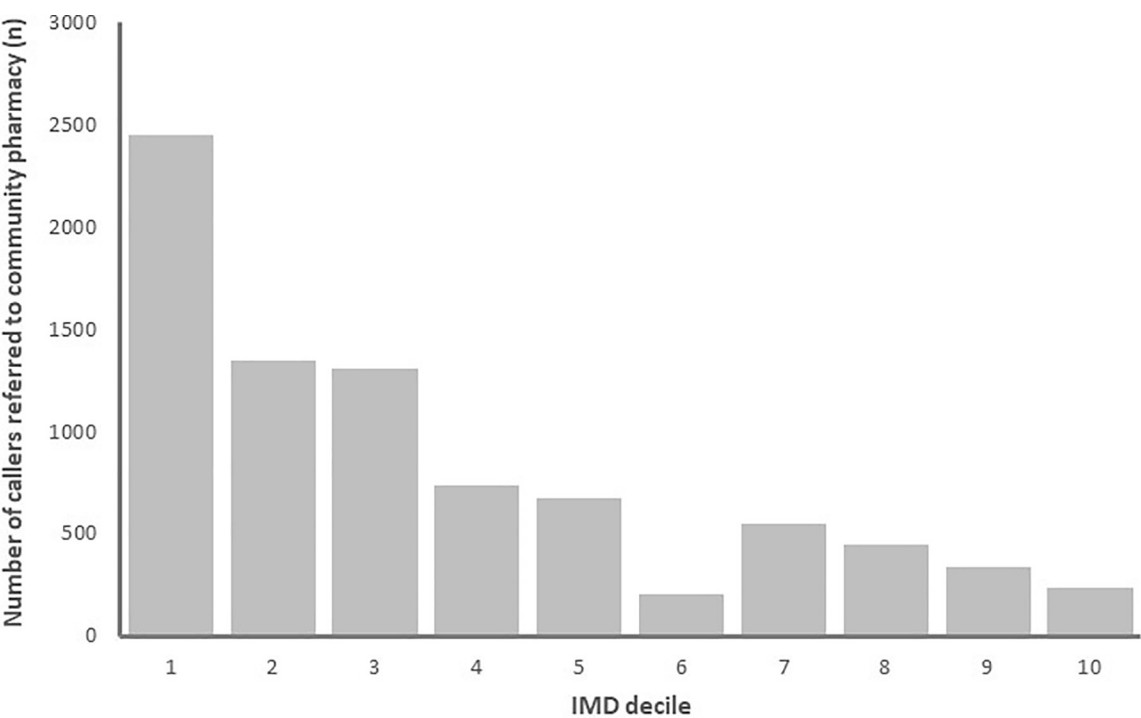

**Fig 5. The number of community pharmacy NHS 111 referrals according to area-level deprivation of the community pharmacy (1 is highest deprivation, 10 is the lowest).**

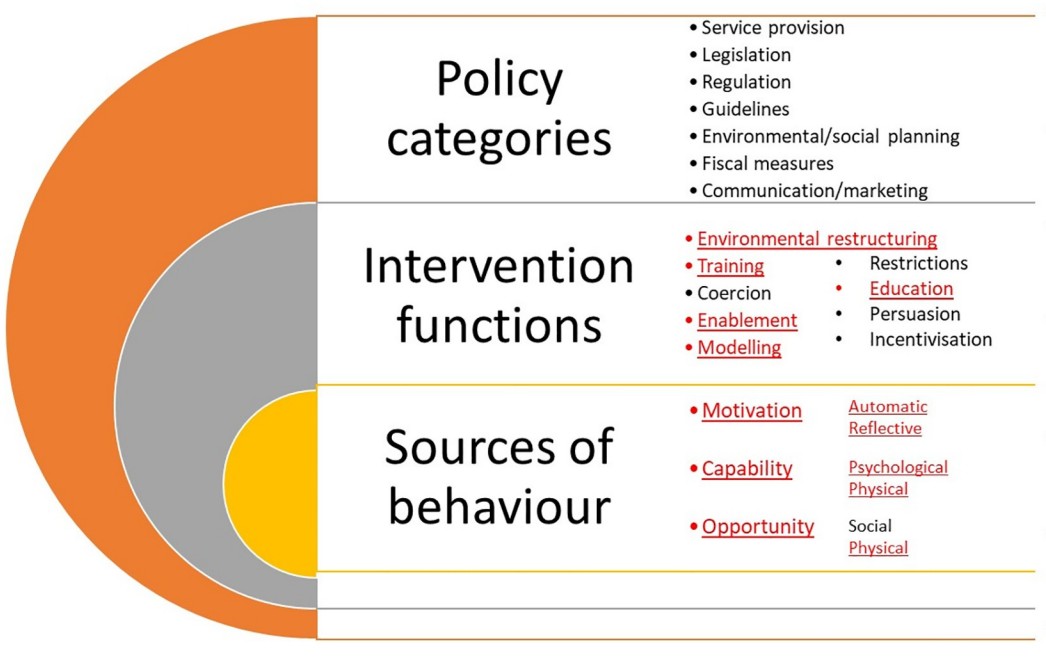

**Fig 6. The behaviour change wheel and the aspects that were referred to, and the emerging themes, through the qualitative interviews with stakeholders.**

opportunity to evidence the contribution of community pharmacy within the wider NHS system, especially in shifting patient burden from other areas of UEC.

> 'It really is an opportunity for pharmacists to demonstrate what they can do for the NHS. . .relieve the significant burden in out of hours services. . .with the evidence, it can really pave the way for more clinical services going to pharmacy.'[TM4]

There was expressed anticipation that DMIRS would improve wider confidence and awareness in community pharmacy, enabling investment into further service optimisation, and development and better integration into the NHS.

- Reflective Motivation of service designers, managers and providers through Education and Enablement

The multi-disciplinary approach taken within the planning workshops that informed the development of DMIRS was appreciated. These activities were crucial to raise awareness and understanding of the interfaces and complexity between NHS 111, UEC, community pharmacy and the patients and public. DMIRS was built upon dialogue and knowledge exchange across these interfaces, and the perceived result was a cohesive service, providing safe and effective care.

> 'The way it was done from the outset. . .the planning, the conversations, it really made a difference. The groundwork done has meant we all have a better understanding of the system. We work better and can see how to move this forward. We haven't done this thorough work for other services before and it shows in the dramatic results this service has achieved.'[TM2]

It was recognised that engagement of service managers was integral to successful implementation and accelerated adoption of DMIRS. Giving the service providers direct access to the managers through engagement events and one-on-one interactions, provided the opportunity to educate them about the wider service system and other actors within the provision. This increase in awareness of the component parts, individual contributions and potential outcomes, resonated strongly with project team members and NHS 111 call members. Linked to this was a perceived increase in the reflective motivation and psychological capability of service providers to engage with service provision.

> 'The enthusiasm from the project lead to come in and talk about this with passion. . .taking time to dedicate to engagement, has really been a significant driver for this service. It has really accelerated change.'[TM6]

- Moving past barriers: Motivation of pharmacists and of patients improved through Education, Modelling and Enablement

Project team members agreed that raising public awareness about the service would create a proactive message about accessing care from the community pharmacy for low acuity conditions. Also further communication and tailored training for service providers, especially community pharmacists, was suggested. For them, increasing reflective motivation was seen as the focus of this promotion, to encourage behaviour change in order to increase engagement and drive for service provision. DMIRS, was deemed vital to not only sustainability and effectiveness of this service, but to improving the integration into the NHS and provision of clinical services.

*'Community pharmacists really need to understand the bigger picture of where this service fits. They need to appreciate how much of an opportunity DMIRS presents. I think that will drive motivation and engagement. . .but communicating to such a large group is difficult. We need more engagement events. . .better attended. . .more marketing.'[TM1]*

## Community pharmacists

- DMIRS benefits from existing Motivation and Capability of pharmacists

Community pharmacists agreed that DMIRS was a good idea and should involve pharmacy. For them, DMIRS was a valuable way to ease pressures on GPs and hospitals. There was a shared sense of a natural fit into the normal workflow of the pharmacy without creating additional burden on pharmacist time, although some felt the associated paperwork was lengthy. It was also seen as an opportunity to make their knowledge and skills known.

*'[Pharmacists are] accessible. . .no appointment needed so should be the first port of call. . .DMIRS is the beginning of the future' [CP4]*

Community pharmacies felt prepared to include more clinical services into their practices. Availability of private consultation rooms, with computers for the referral software was an element of this readiness. Pharmacists found patients quite receptive of the service. The patient response was overall positive, except where the patient required onward referral. This was seen as a standard practice for any patient whose condition might require more specialist attention.

*'I think they [patients] are happy, they seem happy. Except. . .those ones who don't get what they need in the pharmacy and need to go on somewhere else.' [CP10]*

- Improving pharmacists Capability with Education, Environmental restructuring and Enablement

Some pharmacists were concerned about time the consultations took. They felt unable to participate as fully as they had hoped to, but were supportive of DMIRS and wished to be involved as staffing issues were resolved. Further training and upskilling of pharmacists was also identified as an issue. There was a need to be better equipped for managing referrals, and providing the confidence and competence aligning to the increasingly clinical direction of the profession.

*'. . .we can't do chest examinations, and look into people's throats or ears. So some tailored training would be great to get us up to speed.' [CP14]*

*'A lot of what is needed is a good, thorough consultation. We don't need to examine everything, but we do need to do a clinical consultation. . .in most cases this is enough and some of us [community pharmacists] will have the confidence to do this, but I can imagine there a load of pharmacists who might not.'[CP10]*

Some participants felt that a few of the referrals into pharmacy were inappropriate and required escalation to a doctor. Others perceived this as part of a normal care pathway, where further clinical triage in the pharmacy, after a non-clinical telephone conversation, may lead to an onward referral. Some pharmacists acknowledged that the patients' answers to NHS 111

call handlers may not fully reflect the nature of their complaint, resulting in the subsequent escalation from the community pharmacist.

*'Patients may answer questions differently to NHS 111, and we see them* [in the community pharmacy] *it's obvious they should not be here. . ..but we don't know those questions and answers' [CP5]*

Pharmacists mentioned other referral services (e.g. the NHS Urgent Medicines Supply Advanced Service) and felt that it would be beneficial for consistency of care to have an integrated approach to delivery and funding. Moving forward, community pharmacists think DMIRS is viable, deliverable and facilitates delivery of other clinical services.

*'Another step towards educating the public. . .it enables us to provide other services as well-. . .and people, once they have been through this service, will come again to see us.'[CP6]*

- Improving community pharmacy's Opportunity to be involved through better Marketing

There were concerns regarding the slow uptake of the service by the public, and comments about the need to promote DMIRS to manage patient expectations better. Some pharmacists thought the NHS 111 call handlers could educate patients about the role of the pharmacist. Better understanding was thought to help with patients' expectations and reduce inappropriate referrals to community pharmacy.

*'When patients are referred they don't understand what they are going to receive in the pharmacy and don't like having to pay for products' [CP9]*

## Patients and the public

Overall the patients were positive about the service. Only two patients reported poor experiences, both had called with concerns about a young child. They had presented to the referred community pharmacy and were advised to attend the walk-in centre. The patient representatives expressed dissatisfaction and felt they should have originally been referred to the more urgent clinical setting.

*'The pharmacist just said they had to refer us to the walk in centre because they couldn't do anything. It was just a waste of time. We should have got sent there in the first place.'[PP5]*

Most of the patients described their experience of community pharmacy positively. For many, their health concern had been addressed solely with advice or with the addition of an over-the-counter remedy. Patients' previous experience with a community pharmacy prepared them to some degree of the capability and knowledge of the pharmacist.

*'I use my pharmacy all the time for my medication, and they are always really good. So I was happy when the lady on the phone said try the pharmacy first.'[PP11]*

Even if patients' conditions had not been resolved or improved by attending the community pharmacy, they were complementary. Pharmacist were seen as professional and considerate, providing useful reassurance and advice.

*'I showed him the burn on my hand. He gave it a good clean and gave me a dressing but said I would need someone to look at it because of my old skin. He didn't say that, but he said my*

*skin would take longer to heal, especially because of the medication I was on. When I went to the doctor, he said the pharmacist was right and it's good he told me to come because I needed to take some antibiotics for a while after that to stop an infection.'[PP3]*

Those with no previous experience with a community pharmacy, described a sense of surprise and appreciation for their role and the service provided.

*'I didn't know pharmacists knew like how to check you like a doctor or nurse. They did my blood pressure as well, so yea I was happy with that.'[PP8]*

Participants acknowledged the burden on the wider NHS and felt that community pharmacy could provide some relief for patients. For them, community pharmacy provided a good 'middle ground', when the condition needed some attention but not urgent care before a potentially long-awaited GP appointment.

*'I knew I needed someone to have a look but I knew that it wasn't urgent, so I like that the pharmacist could have a look and give advice because it just wasn't like overly urgent or dangerous.'[PP21]*

There was expression that the service and advice that can be provided in a community pharmacy was reassuring, convenient and of clinical value. A few patients felt that if pharmacists had the capacity to do more, like prescribe, the offering would be much more significant.

*'It would be better if a pharmacist could do more. Like she told me I needed antibiotics, but it would have been great if she could have just prescribed some there and then.'[PP17]*

Some patients stated that the experience of attending a community pharmacy *via* DMIRS meant that they will attend a community pharmacy first upon experiencing a minor condition in future rather than contacting NHS 111.

*'Even though I had to go on and see my doctor cos the tablets didn't help very much, I would go back to the pharmacy for something minor. They were just really efficient and listened to me.'[PP28]*

## Discussion

In this study, the quantitative service activity data of the regional minor illness pilot, DMIRS, (service research) has been presented and supplemented with contextual information about the feasibility in the implementation, adoption, delivery and outcomes of that service (implementation research).

The use of this service appears to mirror use of NHS 111 in general, where calls are more likely to be received from females and for younger age groups. [19] Anderson *et al.* also reported a significant volume of calls to NHS 111 were for patients up to the age of 9 years old, but also patients aged over 80 years were highly represented. [20] It is also clear that the pilot has demonstrated the feasibility of shifting patient burden from urgent and emergency care to community pharmacy for a range of low acuity conditions. This corresponds with a previous study which demonstrated that 27.8% of NHS 111 call handler cases referred to Accident and Emergency reviewed by a panel of GPs during Sep-Dec 2014, would have been advised to self-care or other management services (including referral to community pharmacy). [21]

The analysis of the patients being referred to pharmacy, offers service managers and providers the opportunity to optimise resource and training for the future. For example, awareness around the patient groups (demographics) and conditions can inform the educational training packages required for community pharmacists to improve effective service delivery. For stakeholders monitoring the key performance indicators and patient safety; the rates of completion, escalation and rejection are important towards refining the service specification and mitigating risk. Those conditions which currently have high escalation rates, should be considered in service optimisation, but also demonstrate a need for training and capacity in community pharmacy. Many of those conditions with higher escalation rates are infections, which could, after appropriate clinical examination, warrant the need for a prescribed course of antibiotics. Community pharmacists and NHS 111 call handlers have also recognised the potential increased value of DMIRS if more pharmacists were able to safely and appropriately prescribe or supply (e.g. under a patient group direction) antibiotics.

Further valuable deductions possible with this data set relate to the most used, or appropriate medicines for the most prevalent low acuity conditions. Until now a formulary for minor ailments services has not been empirically informed or evidenced, instead included medicines were subject to regional variation and unstandardized decision-making of local Clinical Commissioning Groups. The data from DMIRS offers commissioners and service designers substantial evidence to inform formularies if commissioned minor ailment services were to continue.

The qualitative component of this investigation has provided some insight to potentially explain how a regional pilot gained such traction for wider adoption and, relatively quickly, informed changes to contractual requirements and national practice. Some of the data included in this study was included in an interim pilot evaluation, undertaken by the PhIF evaluation team in NHS England. Evidence from this investigation went on to inform the five-year framework for GP services, published in January 2019, by NHS England. NHS England stated in this document that DMIRS had demonstrated that many patients, who would have otherwise been advised to attend general practice, could have been successfully managed in a community pharmacy setting. [21]

The momentum of interest in the value of DMIRS has been further manifested in the new Community Pharmacy Contractual Framework announced in July 2019. [22] A new 'Community Pharmacy Consultation Service' (CPCS) has been proposed to replace the local pilots of DMIRS, as well the current NHS urgent Medicine Supply Advanced Service (NUMSAS). The new CPCS was proposed to launch in October 2019 as an integrated part of the contractual framework. [22] This means that pharmacists signing up to provide this service would be remunerated as a standard for their provision.

Stakeholders from the NHS 111 system in our study have described behaviour change in their practice that was facilitated through implementation processes influencing various aspects of the policy categories and intervention functions outlined in the behavioural change wheel. [19] The implementation process appears to have impacted these aspects within the different settings that make up the NHS 111 system, e.g. NHS 111 call handler setting, community pharmacy setting. This multi-faceted approach to implementation could be considered a reasonable explanation for the wider adoption and impact of the pilot, since the process of implementation undertook a more holistic, systems approach in setting up and managing service delivery. This strategy also attains a level of success across three concepts of sustainability as outlined by Moullin *et al.*: [23]

- Firstly, the implementation process has fortified the routinisation of pharmacy minor illness service delivery; reinforced the institutionalisation through evolving supportive conditions,

and ensured the maintenance of benefits through adopting technological systems enabling service data collection to evidence service outcomes.

- The implementation process has considered the context and environment rather than just focusing on the specific setting of service delivery, i.e. community pharmacy. The process has influenced factors to ensure the environment is conducive to successful service delivery.

- There was local buy-in and support across the stakeholders involved in the service system as demonstrated in the initial consensus workshops utilised to frame the service. [10] Progressing forward, the national political support and revised fiscal structure of the new pharmacy contract further supports longer term implementation and sustainability. [22]

From the community pharmacists' data around unpreparedness, there is potentially more investment required into the training and persuasion to increase motivation and capability to provide DMIRS. In tackling these areas, the fidelity and quality of service provision will be improved, thereby further enhancing impact, outcomes and sustainability of DMIRS. The evidence from successful implementation and service delivery will support the development and implementation of further clinical service provision from community pharmacy. There has been an indication that the next model of care to layer on top of DMIRS, is GP referral to community pharmacy for minor illness. So, with each iterative development, community pharmacy has the opportunity to evidence their contribution to healthcare, optimise skills and knowledge and become a more integrated player within the NHS system, thereby addressing the key barriers that were identified by the NHS England independent review of community pharmacy clinical services. [9]

The patients and public generally reported positive experience through accessing services *via* DMIRS. Porteous *et al.* also reported that patients preferred to practice self-care for minor conditions and access a community pharmacy for advice with a key motivating factor being reduced waiting times and incurred costs. [24] Todd *et al.* demonstrate that the majority of the population can access a community pharmacy within a 20 minute walk, where no appointment is required. Also, patients in areas of greatest deprivation, which contribute a greater volume of DMIRS referrals, have even greater access to community pharmacy. [25]

The overall long-term aim of DMIRS, and indeed a proposed GP referral to community pharmacy, is to educate and empower patients and the public to promote self-care of minor conditions. Through remodelling the NHS care pathways, highlighting through explicit referral and marketing of community pharmacy, it is anticipated that health-seeking behaviours can be changed to reduce the unsustainable pressures on GP and urgent and emergency care for conditions that patients could manage themselves with minimal clinical input. In order to assess for this as a consequence of DMIRS (and the subsequent CPCS), a longitudinal assessment of patient knowledge and health-seeking practices around low acuity conditions would be pragmatic. It would be anticipated that call volumes for such presenting complaints would also decrease over time, as patients instead self-present at community pharmacy, with no prompting from contact with an NHS service.

## Limitations

Our study is limited by the nature of the routine service activity data made available as part of a service evaluation. It would have been ideal to verify that those patients referred to pharmacy did not re-enter the system by seeking medical care from their GP practice or accessing UEC again. However, tracking patients fell outside the remit and ethical permissions of this work. Similarly, we have explained the exploratory approach we have adopted here to investigate the feasibility of shifting patient burden and scrutinising the implementation process and delivery.

We also propose that an explanatory study is now conducted to investigate the referral rate of patients by NHS 111 call handlers (or fidelity of service provision) to assess the burden of patients still be managed in UEC, and explore the reasons why, e.g. negotiating skills of NHS 111 call handlers. Similarly, completion rate of community pharmacists warrant further study to address any barriers or concerns downstream in the service system that risk successful normalisation and sustainability.

The qualitative data, by its very nature, does not suggest generalisability, however, we have, through adopting inductive thematic saturation, provided some detailed insight into the mechanics of the service implementation and delivery. We have not presented feedback and perspectives from doctors, however, it has been reported in an internal NHS England report, that DMIRS has saved substantial GP appointment time across the pilot period. Also, doctors, specifically those with roles in UEC, were involved in the participatory design of the service as described in our previous study. [10] However, as this service is rolled out nationally and with plans to design and implement a GP to community pharmacy referral pathway for low acuity conditions, engaging with and investigating perspectives of doctors would be informative.

## Conclusions

This study is timely to report upon the service activity data of a regional pilot that rapidly informed healthcare policy and practice in relation to managing the burden of low acuity conditions presenting in the urgent and emergency care setting. The qualitative data provides in-depth, contextual information to highlight the processual underpinnings of the design, implementation and delivery of the service which contributed to its wider adoption. DMIRS has demonstrated to be a success story for community pharmacy as it serves as a step towards improved integration within the wider NHS and provides a proof of concept for the delivery of clinical services with the ability to gather self-certifying evidence of safe, valuable and effective practice.

## Supporting information

**S1 Table. The Template for Intervention Description and Replication (TIDIER) checklist for the digital minor illness referral service.**
(DOCX)

**S1 File. The service activity data for the digital minor illness referral service.**
(XLSM)

**S2 File. Stakeholder semi-structured interview topic guides.**
(DOCX)

## Author Contributions

**Conceptualization:** Hamde Nazar, Andre Yeung.

**Formal analysis:** Hamde Nazar, Cerys Evans, Nicole Kyei, Laura Lindsey, Zachariah Nazar, Katie Thomson, Adam Todd.

**Funding acquisition:** Hamde Nazar.

**Investigation:** Hamde Nazar.

**Methodology:** Hamde Nazar, Laura Lindsey, Zachariah Nazar, Katie Thomson, Adam Todd.

**Project administration:** Hamde Nazar, Andre Yeung.

**Resources:** Hamde Nazar.

**Writing – original draft:** Hamde Nazar.

**Writing – review & editing:** Hamde Nazar, Cerys Evans, Nicole Kyei, Laura Lindsey, Zachariah Nazar, Katie Thomson, Andre Yeung, Adam Todd.

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
