## [Decision Letter · Decision Letter 0]

27 Jan 2020

PONE-D-19-33807

A service evaluation and stakeholder perspectives of an innovative digital minor illness referral service from NHS 111 to community pharmacy

PLOS ONE

Dear Dr. Nazar,

Thank you for submitting your manuscript to PLOS ONE. After careful consideration, we feel that it has merit but does not fully meet PLOS ONE’s publication criteria as it currently stands. Therefore, we invite you to submit a revised version of the manuscript that addresses the points raised during the review process.

In addition to the comments raised by the reviewers and cited below, please also consider the following issues:

The methods refer to the qualitative analysis as using both a modified grounded theory approach as well as thematic analysis.  This is confusing -since no theory was proposed, how is this grounded theory.  Referring to the study as thematic analysis seems suitable.Can the authors comment on a further aspect of pharmacists' opinions - namely that apparently pharmacists were providing referral services without recompense for their time and effort.  The pharmacy literature is replete with "wonderful" studies of pharmacists giving away their services for free.  Such pharmacy programs rarely seem to be scalable or replicable because not all pharmacists are so public minded as to work for free towards the public good.A major stakeholder not included in the analysis is that of physicians.  Can the authors address this oversight.

We would appreciate receiving your revised manuscript by February 15, 2020. To enhance the reproducibility of your results, we recommend that if applicable you deposit your laboratory protocols in protocols.io, where a protocol can be assigned its own identifier (DOI) such that it can be cited independently in the future. For instructions see: http://journals.plos.org/plosone/s/submission-guidelines#loc-laboratory-protocols

We look forward to receiving your revised manuscript.

Kind regards,

John Rovers, PharmD, MIPH

Academic Editor

PLOS ONE

Journal Requirements:

2. Please address the following:

- Please ensure you have thoroughly discussed any potential limitations of this study within the Discussion section, including the potential bias introduced by the sampling method used and the nature of qualitative data.

- Please include additional information regarding the semi-structured interview guide used in the study and ensure that you have provided sufficient details that others could replicate the analyses. For instance, if you developed a guide as part of this study and it is not under a copyright more restrictive than CC-BY, please include a copy, in both the original language and English, as Supporting Information. In addition, please include any details of pre-testing of this guide, including the number of participants and where they were recruited from.

Thank you for your attention to these queries.

"HN received funds to undertake evaluative work for the Digital Minor Illness Referral Service from NHS England (Grant number: BH181784). Funders supported the study design, data collection, analysis and decision to publish the work."

We note that one or more of the authors are employed by a commercial company: Local Professional Network.

Reviewers' comments:

Reviewer's Responses to Questions

**Comments to the Author**

1. Is the manuscript technically sound, and do the data support the conclusions?

Reviewer #1: Yes

Reviewer #2: Yes

2. Has the statistical analysis been performed appropriately and rigorously? 

Reviewer #1: No

Reviewer #2: N/A

3. Have the authors made all data underlying the findings in their manuscript fully available?

Reviewer #1: No

Reviewer #2: Yes

4. Is the manuscript presented in an intelligible fashion and written in standard English?

Reviewer #1: Yes

Reviewer #2: Yes

5. Review Comments to the Author

Reviewer #1: The paper focuses on an important issue, addressing a contemporary population health need. The authors used a large dataset to evaluate the effectiveness of a digital referral system towards shifting the burden of urgent and emergency care services by referring patients calling NHS 111 with low acuity conditions to community pharmacists. To improve the readability and application of the findings of this paper, the authors need to make some major revision of the current version.

The Authors need to identify who their target audience is and tailor the presentation of the results to meet their context and needs. The results of the qualitative strand while important is too long. This will be a challenge for easy assimilation by readers who have less time to read a long paper. The authors should consider alternative forms such as visual representing of the results in a figure to accommodate such readers.

There is a need for more discussion of the system to help readers understand its functionality. Authors need to consider unintended consequences of the system especially in their discussion of the findings. How is this accounted for in the development of the system? Does the system consider patient history such as comorbidities, for instance, in the decision on community pharmacist referral? What is the possible harm from the high proportion of referrals among patients aged 0-9 years considering the inability of this age group to extensively self-report their symptoms (themselves or through a third party such as caregivers) given the reliance of the system on symptoms described by patients?

The quantitative strand of the paper was too descriptive and do not present strong empirical evidence to inform policy decisions and the translation of this intervention in other settings. The dataset holds more potentials for an extensive evaluation of the system. The authors need to apply more robust statistical models in understanding the factors associated with the performance of the system. The data on patients’ demography, for instance, could be used to model the behaviour of rejecting referrals for community pharmacist consultations. As a process evaluation paper aiming to optimise the potentials of the DMIRS system, one will expect a detailed investigation to understand why 1/3 of those referred rejected.

With the current analysis, it is unclear whether the DMIRS is actually shifting the burden of low acuity conditions from urgent and emergency care. It is possible for those referred and completed to go back to their GP or A&E. The possibility of this happening was reiterated in the interview result (lines 270-272). The authors need to consider this in plans to improve the statistical analysis.

The Authors reported adopting a mixed-methods design in their paper. Although the results of analyses of qualitative and quantitative data were presented, there was a lack of integration of both methods expected in a mixed-methods design. It is unclear how the design of the interview relates to the analysis and results of the service activity data. Both strands do not seem to complement each other. For instance, the interview result in lines 258-264 on the negotiating abilities of the call handlers could inform statistical investigation towards the level of rejections. Rejection as an outcome of the referral process could be patterned alongside call handler characteristics such as negotiating ability and understanding this pattern is important in optimising acceptance of the referral system.

The qualitative interviews present important insights into the implementation of the intervention and can be a standalone study. I recommend the authors consider splitting the manuscript into two papers, focusing the current paper on the qualitative study exploring the experiences and perspectives of stakeholders in the implementation of the DMIRS system in North East of England. Alternatively, to maintain a mixed-methods design, the statistical analysis should be improved by conducting more robust investigations, and both strands should be well integrated.

Minor comments

Line 51: brief explanation (1 or 2 sentences) of what low acuity conditions mean. This will be useful to the non-clinical audience.

Line 94: is this an exploratory or explanatory mixed methods design? It is unclear how the design of the interview relates to the service activity data. Both strands do not seem to complement each other.

Line 105: add some examples of these specific minor illnesses. It is helpful to prepare your readers at this point on the conditions considered to be low acuity.

Lines 222-226: I was interested in seeing how this will be interpreted in the discussion section. Are those of lower IMD generally more likely to call NHS 111 for low acuity conditions compared to those of higher IMD? Are those of higher IMD more likely to call NHS 111 for more serious/high acuity conditions. Can these explain the high referral into the service seen in the most deprived areas of England?

Line 231: what informed the decision on the number of interviews to conduct and the proportion of each group (CH, TM, CP) contributing to the stakeholders' data? This needs to be explained.

Reviewer #2: Overall, I think this is an interesting evaluation of a unique (and needed) service. (Granted, I am a pharmacist, so I am a bit biased in that area.) The paper is well-written and makes a compelling argument in favor of these types of referrals to community pharmacists for the treatment of minor illnesses. My comments are mainly related to adding additional detail to help readers who are unfamiliar with the UK health care system better understand the referral service developed in this pilot program.

1. At several points in the paper, a list of low acuity conditions are mentioned, but these are never really defined until the Results. Were these conditioned defined prior to the implementation of this service (e.g., as part of the pilot program development)? Some discussion of what these conditions are and how they were selected might be helpful, especially for those who might be interested in implementing this outside the UK.

2. Related to the previous point, some Canadian provinces have implemented pharmacist prescribing for minor ailments. It might be helpful to provide some mention of these efforts outside the UK to provide additional context. This would probably be a discussion point.

3. In the introduction, some of the information struck me as potentially being better suited for the Discussion section. Mainly, the 2nd and 3rd full paragraphs on page 4. Those seem to discuss the status of the DMIRS program after what is being evaluated in the current paper. By presenting these up front, it felt a little out of order to me. You might consider moving these to the end of the paper as a sort of “where is it now” part of the discussion.

4. It would be helpful to explain who the call handlers were. Were these nurses or somebody else? If not nurses, did they have any health care training? That was a big question in my mind in terms of their ability to make appropriate referrals to the community pharmacist vs. to the GP or ED.

5. When referrals were made to a community pharmacy, the call handler selected the pharmacy closest to the patient’s home address. Was there any consideration of whether or not that was the patient’s “usual” pharmacy? Sometimes people use pharmacies that aren’t necessarily the closest geographically to their home.

6. For the participating pharmacies, were these all community pharmacies or only certain ones who were signed up to be part of this pilot service? (My apologies if I missed that in the paper.)

7. On page 6 under Service activity data, the first paragraph and the data extra description seemed a little redundant. You might consider reworking those to combine them without being repetitive. For example, the first paragraph mentions the what was recorded and then the second part notes the data extracted. These are basically the same elements in that prior paragraph.

8. Under Sampling (page 7, line 152), patients attending the community pharmacy referral were asked to provide consent for participation in research and evaluation. Strictly speaking, this project is not research, rather it is an evaluation. Is the “research and evaluation” terminology used because the consent was for the DMIRS pilot more generally (and potential follow-up projects)? You might clarify this and update the wording if needed.

9. In Table 2, the term “safety netting” is used, but it is not defined. This may be a common term for the UK context, but it may not be familiar to those elsewhere.

10. In the results referring to the level of completion (bottom of page 10, line 209; top of page 11, line 214), a percentage is listed in parentheses. It is unclear if these percentages are what were used for “good” vs. “not good” in terms of completion. Please clarify this.

11. The supplemental figure for the Michie wheel (page 12, line 243) was not available when I checked the supplemental information. I was only able to access the TIDieR table. I’m not sure if this was a technological issue on my end or if the file wasn’t attached.

6. PLOS authors have the option to publish the peer review history of their article (what does this mean?). If published, this will include your full peer review and any attached files.

Reviewer #1: Yes: Dr Philip Anyanwu

Reviewer #2: Yes: Spencer E. Harpe

---

## [Author Response · Author response to Decision Letter 0]

5 Feb 2020

Response to editor:

These style changes have been made.

2. Please address the following:

- Please ensure you have thoroughly discussed any potential limitations of this study within the Discussion section, including the potential bias introduced by the sampling method used and the nature of qualitative data.

Response to editor:

Thank you for highlighting this missing information which has now been included.

- Please include additional information regarding the semi-structured interview guide used in the study and ensure that you have provided sufficient details that others could replicate the analyses. For instance, if you developed a guide as part of this study and it is not under a copyright more restrictive than CC-BY, please include a copy, in both the original language and English, as Supporting Information. In addition, please include any details of pre-testing of this guide, including the number of participants and where they were recruited from.

Thank you for your attention to these queries.

Response to editor:

Apologies, we intended to supply the interviews guides but appear not have done so with the original submission. We have made them available in this submission.

"HN received funds to undertake evaluative work for the Digital Minor Illness Referral Service from NHS England (Grant number: BH181784). Funders supported the study design, data collection, analysis and decision to publish the work."

We note that one or more of the authors are employed by a commercial company: Local Professional Network.

“The funder provided support in the form of salaries for authors [HN], but did not have any additional role in the study design, data collection and analysis, decision to publish, or preparation of the manuscript. The specific roles of these authors are articulated in the ‘author contributions’ section.”

Response to editor

We have included an amended financial and competing interest statement in our cover letter as requested.

Response to editor:

We now include the data set as supporting material.

Response to editor:

These changes have been made.

The methods refer to the qualitative analysis as using both a modified grounded theory approach as well as thematic analysis. This is confusing -since no theory was proposed, how is this grounded theory. Referring to the study as thematic analysis seems suitable.

Response to editor:

Thank you for highlighting this discrepancy. We have addressed this by indicating a pragmatic convergent (or concurrent) design to data collection was adopted with an inductive coding strategy leading to thematic analysis. We believe this more accurately describes our data collection and analysis.

Can the authors comment on a further aspect of pharmacists' opinions - namely that apparently pharmacists were providing referral services without recompense for their time and effort. The pharmacy literature is replete with "wonderful" studies of pharmacists giving away their services for free. Such pharmacy programs rarely seem to be scalable or replicable because not all pharmacists are so public minded as to work for free towards the public good.

Response to editor:

I appreciate this perspective, however this pilot led to a change to the national pharmacy contract, so that this service is now subject to remuneration. So we do not present a case where pharmacists are doing this for free, instead this study has provided evidence to demonstrate pharmacists role in minor ailment management (which before now has not been tangible) to substantiate the need to remunerate pharmacists. 

A major stakeholder not included in the analysis is that of physicians. Can the authors address this oversight.

Response to editor:

Thank you, yes I see why this could be seen as an oversight, however, doctors (specifically out-of-hours doctors) were involved in the initial design of the service. Their perspectives were taken into account about management and safety of the patients. It would be useful to now interview doctors about their views of a wider roll out of this service which we have recommended for future work.

Reviewer #1: 

The paper focuses on an important issue, addressing a contemporary population health need. The authors used a large dataset to evaluate the effectiveness of a digital referral system towards shifting the burden of urgent and emergency care services by referring patients calling NHS 111 with low acuity conditions to community pharmacists. To improve the readability and application of the findings of this paper, the authors need to make some major revision of the current version.

Response to reviewer:

We thank the reviewer for his thorough considerations and suggestions for the work and hope amendments have made the work more readable and comprehensive.

The Authors need to identify who their target audience is and tailor the presentation of the results to meet their context and needs. The results of the qualitative strand while important is too long. This will be a challenge for easy assimilation by readers who have less time to read a long paper. The authors should consider alternative forms such as visual representing of the results in a figure to accommodate such readers.

Response to reviewer:

We have introduced the aim of the work in the ‘study design’, where we outline how the study design has been tailored to the interest of service designers, commissioners and potentially policy makers. We concede that the qualitative data, as might be expected, is lengthy, we have developed a visual representation that we hope summarises much of the findings to accommodate reads wanting a more efficient means of understanding the data.

There is a need for more discussion of the system to help readers understand its functionality. Authors need to consider unintended consequences of the system especially in their discussion of the findings. How is this accounted for in the development of the system? Does the system consider patient history such as comorbidities, for instance, in the decision on community pharmacist referral? What is the possible harm from the high proportion of referrals among patients aged 0-9 years considering the inability of this age group to extensively self-report their symptoms (themselves or through a third party such as caregivers) given the reliance of the system on symptoms described by patients?

Response to reviewer:

We have included figure 1, from a previous study, to better represent the NHS 111 system. We appreciate the questions posed by the reviewer as crucial in the service design. These considerations have been addressed in our previous study, were a participatory design approach was utilised to account for pharmacist competence, availability of resources and treatments in the pharmacy and managing potential patient escalation or onward referral. 

The quantitative strand of the paper was too descriptive and do not present strong empirical evidence to inform policy decisions and the translation of this intervention in other settings. The dataset holds more potentials for an extensive evaluation of the system. The authors need to apply more robust statistical models in understanding the factors associated with the performance of the system. The data on patients’ demography, for instance, could be used to model the behaviour of rejecting referrals for community pharmacist consultations. As a process evaluation paper aiming to optimise the potentials of the DMIRS system, one will expect a detailed investigation to understand why 1/3 of those referred rejected.

Response to reviewer:

These are very valid and useful comments. In reframing our study as a pilot, assessing the feasibility in implementing and delivering this service, we believe the descriptive nature of the quantitative service activity data is appropriate. The service activity data has provided evidence to policy-makers that referral to community pharmacy is a feasible pathway. Our study aim was to assess the feasibility of delivering the service, rather than explaining the service or exploring factors associated with how well it performs. Data not collected by the research team, has allowed the same policy makers to assess that the new service pathway saves GP time, and does not result in duplicated presentation of patients at GP practices and out-of-hours services. Unfortunately, as this data was not part of the routine service activity data, it has not been made accessible to the research team to complement and substantiate the findings in this work.

With the current analysis, it is unclear whether the DMIRS is actually shifting the burden of low acuity conditions from urgent and emergency care. It is possible for those referred and completed to go back to their GP or A&E. The possibility of this happening was reiterated in the interview result (lines 270-272). The authors need to consider this in plans to improve the statistical analysis.

Response to reviewer:

An in-depth process evaluation to understand some of the service activity figures is indeed now warranted, given that the pathway has shown to be feasible. This is outside the scope and the aim of this study and would recommend current service leads and commissioners to monitor and evaluate service performance to understand the service outcomes. This would ensure cycles of quality improvement towards more effective and efficient service delivery. As reiterated previously, service leads are reassured that patients are not re-entering the system. However, given this relies on access to patient-identifiable data to allow trackability, the research team did not have access to this information to assess and therefore report it here.

The Authors reported adopting a mixed-methods design in their paper. Although the results of analyses of qualitative and quantitative data were presented, there was a lack of integration of both methods expected in a mixed-methods design. It is unclear how the design of the interview relates to the analysis and results of the service activity data. Both strands do not seem to complement each other. For instance, the interview result in lines 258-264 on the negotiating abilities of the call handlers could inform statistical investigation towards the level of rejections. Rejection as an outcome of the referral process could be patterned alongside call handler characteristics such as negotiating ability and understanding this pattern is important in optimising acceptance of the referral system.

Response to reviewer:

We appreciate the reviewer’s comments about the perceived lack of coherency between the adopted methodologies. We hope that the reframing of the paper around investigating feasibility of the referral pathway addresses this concern. We explain our convergent study design and how integration of findings were undertaken post data collection to scrutinise the feasibility and outcome of the service. In relation to better understanding the presented service activity data, we agree with the reviewer, that an explanatory approach is appropriate and potentially prove very useful. In our exploratory approach, we would contest, that we have identified potential aspects of the system and the process would warrant evaluation going forward, e.g. observational assessment of call handlers as they manage patient calls to assess the correlation between negotiating capabilities and referral uptake rate. We have included this proposition in our recommendations for future work.

The qualitative interviews present important insights into the implementation of the intervention and can be a standalone study. I recommend the authors consider splitting the manuscript into two papers, focusing the current paper on the qualitative study exploring the experiences and perspectives of stakeholders in the implementation of the DMIRS system in North East of England. Alternatively, to maintain a mixed-methods design, the statistical analysis should be improved by conducting more robust investigations, and both strands should be well integrated.

Response to reviewer:

We appreciate this recommendation, but would reflect that such a study would be one with a different set of aims and objectives; a set that looks to investigate in-depth the nuances of service outcomes and effects. This would be a much needed study to follow-on from and be informed by the work we present in this paper. 

We believe that coupling the service activity data alongside the qualitative data provides a more informed perspective of feasibility and better highlights aspects of the service that require further scrutiny and evaluation.

Minor comments

Line 51: brief explanation (1 or 2 sentences) of what low acuity conditions mean. This will be useful to the non-clinical audience.

Response to reviewer:

We have added in an explanation so the reader is made aware at the onset of the paper.

Line 94: is this an exploratory or explanatory mixed methods design? It is unclear how the design of the interview relates to the service activity data. Both strands do not seem to complement each other.

Response to reviewer:

We thank the reviewer for highlighting this and have added some text to explain the exploratory approach adopted here.

Line 105: add some examples of these specific minor illnesses. It is helpful to prepare your readers at this point on the conditions considered to be low acuity.

Response to reviewer:

We have added a few examples to raise awareness earlier on reading.

Lines 222-226: I was interested in seeing how this will be interpreted in the discussion section. Are those of lower IMD generally more likely to call NHS 111 for low acuity conditions compared to those of higher IMD? Are those of higher IMD more likely to call NHS 111 for more serious/high acuity conditions. Can these explain the high referral into the service seen in the most deprived areas of England?

Response to reviewer:

Indeed those are very interesting lines of inquiry, however, we only have data for patients referred into DMIRS, so cannot make assumptions of all patients calling NHS 111 for low acuity conditions. This would warrant further investigation. 

Line 231: what informed the decision on the number of interviews to conduct and the proportion of each group (CH, TM, CP) contributing to the stakeholders' data? This needs to be explained.

Response to reviewer:

We have included our approach to continue data collection until inductive thematic saturation had been reached. This guided the number of interviews conducted within each stakeholder group.

Reviewer #2: Overall, I think this is an interesting evaluation of a unique (and needed) service. (Granted, I am a pharmacist, so I am a bit biased in that area.) The paper is well-written and makes a compelling argument in favor of these types of referrals to community pharmacists for the treatment of minor illnesses. My comments are mainly related to adding additional detail to help readers who are unfamiliar with the UK health care system better understand the referral service developed in this pilot program.

Response to reviewer:

We thank the reviewer for their considerate and thorough review of our work.

1. At several points in the paper, a list of low acuity conditions are mentioned, but these are never really defined until the Results. Were these conditioned defined prior to the implementation of this service (e.g., as part of the pilot program development)? Some discussion of what these conditions are and how they were selected might be helpful, especially for those who might be interested in implementing this outside the UK.

Response to reviewer:

With thanks, we have provided an explanation of minor conditions and some examples in the introduction so readers are made aware early on in reading the paper.

2. Related to the previous point, some Canadian provinces have implemented pharmacist prescribing for minor ailments. It might be helpful to provide some mention of these efforts outside the UK to provide additional context. This would probably be a discussion point.

Response to reviewer:

We have made mention that a recent systematic review investigated minor ailment services internationally and found studies published from UK and Canada so readers are made aware of thwe wider provision of these services.

3. In the introduction, some of the information struck me as potentially being better suited for the Discussion section. Mainly, the 2nd and 3rd full paragraphs on page 4. Those seem to discuss the status of the DMIRS program after what is being evaluated in the current paper. By presenting these up front, it felt a little out of order to me. You might consider moving these to the end of the paper as a sort of “where is it now” part of the discussion.

Response to reviewer:

We appreciate this perspective and have moved this information to the discussion as suggested. We agree that this is more appropriate as data presented in the paper did lead to those specific developments.

4. It would be helpful to explain who the call handlers were. Were these nurses or somebody else? If not nurses, did they have any health care training? That was a big question in my mind in terms of their ability to make appropriate referrals to the community pharmacist vs. to the GP or ED.

Response to reviewer:

We have verified that call handlers are not health-care trained staff.

5. When referrals were made to a community pharmacy, the call handler selected the pharmacy closest to the patient’s home address. Was there any consideration of whether or not that was the patient’s “usual” pharmacy? Sometimes people use pharmacies that aren’t necessarily the closest geographically to their home.

Response to reviewer:

Indeed we can agree this is the case. However, other factors were taken into consideration when selecting the pharmacy: opening hours (given that many patients called outside of normal working hours), and sending to only those pharmacies which had signed up to provide the service. This has been clarified in the text.

6. For the participating pharmacies, were these all community pharmacies or only certain ones who were signed up to be part of this pilot service? (My apologies if I missed that in the paper.)

Response to reviewer:

Apologies, this was missing from the paper and has now been added.

7. On page 6 under Service activity data, the first paragraph and the data extra description seemed a little redundant. You might consider reworking those to combine them without being repetitive. For example, the first paragraph mentions the what was recorded and then the second part notes the data extracted. These are basically the same elements in that prior paragraph.

Response to reviewer:

Thank you for pointing this out. We have moved repetitious wording as recommended.

8. Under Sampling (page 7, line 152), patients attending the community pharmacy referral were asked to provide consent for participation in research and evaluation. Strictly speaking, this project is not research, rather it is an evaluation. Is the “research and evaluation” terminology used because the consent was for the DMIRS pilot more generally (and potential follow-up projects)? You might clarify this and update the wording if needed.

Response to reviewer:

Apologies, yes this could be misleading. We have removed the ‘research’ component as this was indeed considered service evaluation.

9. In Table 2, the term “safety netting” is used, but it is not defined. This may be a common term for the UK context, but it may not be familiar to those elsewhere.

Response to reviewer:

Yes, this is an oversight on our part and we have now provided some further explanation.

10. In the results referring to the level of completion (bottom of page 10, line 209; top of page 11, line 214), a percentage is listed in parentheses. It is unclear if these percentages are what were used for “good” vs. “not good” in terms of completion. Please clarify this.

Response to reviewer:

Yes, the percentages in the parentheses were misplaced in the sentence making it confusing; this has been amended. 

11. The supplemental figure for the Michie wheel (page 12, line 243) was not available when I checked the supplemental information. I was only able to access the TIDieR table. I’m not sure if this was a technological issue on my end or if the file wasn’t attached.

Response to reviewer:

Apologies, this was missing. We have now incorporated this figure into the manuscript to visually summarise the qualitative data (figure 6).

---

## [Decision Letter · Decision Letter 1]

27 Feb 2020

A service evaluation and stakeholder perspectives of an innovative digital minor illness referral service from NHS 111 to community pharmacy

PONE-D-19-33807R1

Dear Dr. Nazar,

We are pleased to inform you that your manuscript has been judged scientifically suitable for publication and will be formally accepted for publication once it complies with all outstanding technical requirements.

With kind regards,

John Rovers, PharmD, MIPH

Academic Editor

PLOS ONE

Additional Editor Comments (optional):

Reviewers' comments:

Reviewer's Responses to Questions

**Comments to the Author**

1. If the authors have adequately addressed your comments raised in a previous round of review and you feel that this manuscript is now acceptable for publication, you may indicate that here to bypass the “Comments to the Author” section, enter your conflict of interest statement in the “Confidential to Editor” section, and submit your "Accept" recommendation.

Reviewer #1: All comments have been addressed

Reviewer #2: All comments have been addressed

2. Is the manuscript technically sound, and do the data support the conclusions?

Reviewer #1: Yes

Reviewer #2: Yes

3. Has the statistical analysis been performed appropriately and rigorously? 

Reviewer #1: Yes

Reviewer #2: N/A

4. Have the authors made all data underlying the findings in their manuscript fully available?

Reviewer #1: Yes

Reviewer #2: Yes

5. Is the manuscript presented in an intelligible fashion and written in standard English?

Reviewer #1: Yes

Reviewer #2: Yes

6. Review Comments to the Author

Reviewer #1: Authors have responded to review comments and amended the manuscript accordingly.

A limitation section has been included by the authors to further demonstrate the scope of the investigations conducted.

Reviewer #2: Thank you for your responses to the comments from the editors and the reviewers. All of my concerns were addressed. I think the paper is stronger and tells a more clear story about this interesting service that was implemented.

7. PLOS authors have the option to publish the peer review history of their article (what does this mean?). If published, this will include your full peer review and any attached files.

Reviewer #1: Yes: Dr. Philip Anyanwu

Reviewer #2: Yes: Spencer E. Harpe

---

## [Editor Report · Acceptance letter]

5 Mar 2020

PONE-D-19-33807R1 

A service evaluation and stakeholder perspectives of an innovative digital minor illness referral service from NHS 111 to community pharmacy 

Dear Dr. Nazar:

I am pleased to inform you that your manuscript has been deemed suitable for publication in PLOS ONE. Congratulations! Your manuscript is now with our production department. 

With kind regards,

on behalf of

Dr. John Rovers 

Academic Editor

PLOS ONE